# A Smartcard-Based User-Controlled Single Sign-On for Privacy Preservation in 5G-IoT Telemedicine Systems

**DOI:** 10.3390/s21082880

**Published:** 2021-04-20

**Authors:** Tzu-Wei Lin, Chien-Lung Hsu, Tuan-Vinh Le, Chung-Fu Lu, Bo-Yu Huang

**Affiliations:** 1Graduate Institute of Business and Management, Chang Gung University, Taoyuan 333, Taiwan; d0640001@cgu.edu.tw (T.-W.L.); tvle.cgu@gmail.com (T.-V.L.); 2Department of Information Management, Chang Gung University, Taoyuan 333, Taiwan; hpy53bert@gmail.com; 3Healthy Aging Research Center, Chang Gung University, Taoyuan 333, Taiwan; 4Department of Visual Communication Design, Ming-Chi University of Technology, New Taipei City 243, Taiwan; 5Department of Nursing, Chang Gung Memorial Hospital, Taoyuan 333, Taiwan; 6Department of Information Management, Chihlee University of Technology, New Taipei City 220, Taiwan; peter61@mail.chihlee.edu.tw

**Keywords:** telemedicine systems, user-controlled, single sign-on, multi-server, BAN logic, AVISPA

## Abstract

Healthcare is now an important part of daily life because of rising consciousness of health management. Medical professionals can know users’ health condition if they are able to access information immediately. Telemedicine systems, which provides long distance medical communication and services, is a multi-functional remote medical service that can help patients in bed in long-distance communication environments. As telemedicine systems work in public networks, privacy preservation issue of sensitive and private transmitted information is important. One of the means of proving a user’s identity are user-controlled single sign-on (UCSSO) authentication scheme, which can establish a secure communication channel using authenticated session keys between the users and servers of telemedicine systems, without threats of eavesdropping, impersonation, etc., and allow patients access to multiple telemedicine services with a pair of identity and password. In this paper, we proposed a smartcard-based user-controlled single sign-on (SC-UCSSO) for telemedicine systems that not only remains above merits but achieves privacy preservation and enhances security and performance compared to previous schemes that were proved with BAN logic and automated validation of internet security protocols and applications (AVISPA).

## 1. Introduction

Healthcare is now an important part of daily life because of rising consciousness of health management. People can check up health conditions by themselves, such as heartbeat rate, quality of sleep, amount of exercise, and so on, by supporting wearable technology, including smart phone, smart watch, smart bracelet, etc., which measures biodata and assists self-health management. Currently, biodata is only transferred to a smartphone and analyzed by applications on a smart phone, without being transferred to other outside systems [1]. Medical professionals can know users’ health conditions, if medical professionals are able to access the information immediately [1].

Telemedicine systems provide long distance medical communication and services through which patient and medical professionals can communicate online, and patient benefits from being supported with ambulatory care or other medical services, even in remote areas [1,2,3,4]. Telemedicine systems allow health related data and image to be reliably transmitted from one point to another [1]. Many researchers focused on monitoring patient’s health with specific diseases using telemedicine, such as diabetes and Parkinson’s disease, and telemedicine systems can help a patient recover from illness through this way [5,6,7,8]. In other words, telemedicine can help patients improve their quality of life [1]. Moreover, telemedicine systems can provide better solutions in emergency situations and serious disease monitoring [1,8,9,10]. Telemedicine systems are implemented with wireless communication environments, such as Wi-Fi, Internet of Things (IoT), and fifth generation (5G), to achieve long distance medical communication and services [8,11]. The sensors, such as wearable devices, for example, gather measured data, and measured data are then transmitted through gateways, 5G base stations, and core networks. After this, data is stored or analyzed by applications in back-end servers or cloud servers.

Telemedicine systems are implemented with wireless communication environments, which means data are transmitted through public networks. The patient sends healthcare-related information through public networks when using telemedicine technology, and the transmitted information is important, sensitive, and private [1]. Security issues related to data transmission were discussed, such as eavesdropping, man-in-the-middle (MITM) attack, data tempering attack, message modification attack, data interception attack, etc. [12] Although the Health Insurance Portability and Accountability Act (HIPAA), General Data Protection Regulation (GDPR), and Safe Harbor Laws have regulations that provide privacy of personal information, technical support is still not enough [12,13,14].

Security issues exist in multi-server environments when applying conventional password-based authenticated key exchange schemes. Users have to maintain pairs of identifiers and passwords that increase computational cost and security risks. Moreover, a trustworthy third party is required, while users utilize a single pair of identity and password in multi-server telemedicine systems. However, a malicious third party is able to impersonate users to access other services, with knowledge of shared keys. The same can be proved if a malicious server exists. In terms of performance, the cost of establishing a session key for users is related to the number of servers in conventional schemes. A single sign-on (SSO) mechanism can overcome the above issues that allow users a single action to achieve authentication with a single pair of identity and password, rather than with multiple passwords [15,16,17].

In the proposed scheme, we apply a user-controlled SSO (UCSSO) authenticated key agreement. The key allows patients access to services in multi-server telemedicine systems with a user-defined password, and establishes a secret shared key among servers for securing subsequent communications and designing a smartcard-based user-controlled single sign-on (SC-UCSSO) scheme that can be applied in 5G-IoT multi-server telemedicine systems. The patients has data ownership as they can control and decide the data’s destination and time of transmission. The proposed scheme establishes a secure communication channel using authenticated session keys between patients and services, while meeting general security requirements. Moreover, computational complexity is better than the compared previous schemes. We sketch the remaining organization of paper below. We introduce telemedicine systems and the Chebyshev chaotic maps in Section 2. We introduce our scheme in Section 3, and security and performance analysis are detailed in Section 4 and Section 5. We present our results of implementation in Section 6. Finally, the conclusion is drawn in Section 7.

## 2. Related Works

### 2.1. Telemedicine Systems

Telemedicine systems is a technology of electronic message and telecommunication related to healthcare [1,18]. The National Health Services (NHS) of United Kingdom defines changes to NHS service model as “out of hospital care”, “reducing emergency hospital services”, “personalized care”, “digitally enables care”, and “integrated care systems models”, which can correspond to the features of telemedicine systems [19,20]. Thanks to the IoT technology, which enable medical professionals to monitor patients who are outside of medical institutes in real-time, medical professionals can know users’ health condition, if they are able to view the information immediately [1,20]. In other words, telemedicine systems with IoT can enhance functions of patient’s health monitor and proactive and preventive healthcare interventions [20].

A general telemedicine system can be divided into three level [21]. Level 1 (primary healthcare unit) consists of users with webcam, smart phone, or wearable devices, which is enables communications of measured biodata through wireless communications, including radio frequency identification (RFID), near field communication (NFC), Bluetooth, Wi-Fi, Zigbee, etc. [20]. Measured biodata are transmitted to the user’s smartphone without being transferred to other systems [1]. Level 2 (city or district hospital) is clinic or local hospital that the patient might visit before being transferred to a large hospital or medical center. Level 3 (specialty center) takes part in telemedicine in case of a rare disease or an incurable disease [21]. Figure 1 illustrates a general telemedicine system including two scenarios—asynchronous telemedicine and synchronous telemedicine [21,22]. Asynchronous telemedicine allows patients to decide a proper time to send medical image and health record to medical service providers for detailed examination. Synchronous telemedicine, also called synchronous video conferencing or interactive telemedicine, provides real-time communication between patient and medical professional [22].

### 2.2. Medical Privacy

Telemedicine systems have many challenges, such as infrastructure, connections, professional requirements, data management, and real-time monitoring [23,24]. Medical privacy is of the utmost importance, and damage of medical privacy not only brings huge economic losses and losses of credibility to hospitals and other related institutions but does potential harm to patients and endangers lives of patients [24,25]. Unfortunately, thus far, healthcare-related industries did not achieve users’ expectations [24]. Trust management (TM) is important for allowing reliable data collection and transmission, to provide qualified services and enhance user privacy and information security [26]. Gambetta first defined two widely accepted definitions of trust called reliability trust and decision trust [26,27]. Recently, researchers had discussions about TM of IoT [28,29,30,31,32]. Fortino et al. summarized and discussed main trust concepts, including behavior trust, reputation, honesty, and accuracy [26].

As we mentioned, telemedicine is implemented in public networks, so privacy preservation is one of notable security issues, which has caught researchers’ attention. Mishra et al. [33] and Renuka et al. [34] utilized a biometric feature to design authentication schemes for telemedicine systems. Zriqat and Altamimi discussed issues through data collection, data transmission, and data storage and access level [12]. Dharminder et al. discussed authorized access to healthcare services [35]. Zhang et al. [36], Zhang et al. [37], and Sureshkumar et al. [38] designed authentication and key agreement for telemedicine system. Baker et al. [8], Guo et al. [39], and Anwar et al. [11] focused on telemedicine using IoT, blockchain, and 5G technology and proposed framework or scheme. In summary, three keys to the question must be solved for assuring telemedicine environments. First, image storage should be highly efficient. Second, transmitting sensitive image should satisfy confidence, integrity, and accessibility. Finally, encryption progress should be efficient, especially for the end-point.

## 3. Proposed Scheme

In the proposed system, there are *i* users and *j* servers. User $U_{i}$ can use a smartcard or a smart token to log in to whichever server $S_{j}$ user $U_{i}$ wishes to access, as shown in Figure 2. The proposed scheme includes four phases—system initialization phase, registration phase, authenticated key exchange phase, and offline password change phase. In the system initialization phase, Server $S_{j}$ generates essential parameters and functions for the whole scheme. User $U_{i}$ becomes a legitimate member in the system through the registration phase. In the authenticated key exchange phase, User $U_{i}$ and server $S_{j}$ authenticate each other and establish a session key for symmetric encryption for communication and transmitted measured biodata. The proposed scheme provides offline password change phase such that user $U_{i}$ can change the password periodically, without the participation of server $S_{j}$. Notations are defined in Table 1.

### 3.1. Preliminary

We briefly introduce Chebyshev chaotic maps in this section. The chaotic system has properties that can correspond to the cryptosystem’s properties. First, the result is unpredictable if small changes in initial values occur [40,41,42]. Second, the chaotic system is a complex oscillation [40,41,42]. Third, the chaotic system shows qualitative change of character of solutions [40,41,42]. The above features can correspond to confusion and diffusion of the cryptosystem, which was discussed for decades [24,40,41,42,43,44,45,46,47,48,49,50]. Mathematical definitions of the Chebyshev chaotic maps are introduced as below [24,46,47,48,49,50].

Polynomials of Chebyshev chaotic maps $T_{n}(x):\;[-1,1]\to [-1,1]$ is formed as $T_{n}{(x)=\cos(\mathrm{ncos}}^{-1}\left(x)\right)$ in *x* of degree *n*.If n ≥ 2, polynomials of the Chebyshev chaotic maps is formed as $T_{n}{(x)=2\mathrm{xT}}_{n-1}{(x)-T}_{n-2}\left(x\right)$. However, results of the Chebyshev chaotic maps are 1 and *x* when *n* is 0 and 1, respectively.If (s, r) ∈ Z and s∈[−1,1], $T_{r}{(T}_{s}{(x))=T}_{\mathrm{rs}}{(x)=T}_{s}{(T}_{r}\left(x)\right)$, which is the so-called semi-group property.Zhang [51] proved that semi-group property can hold if Chebyshev polynomials are extended on interval [−∞,+∞]. In the situation, polynomials of Chebyshev chaotic maps become $T_{n}{(x)=(2\mathrm{xT}}_{n-1}{(x)-T}_{n-2}(x))\;\mathrm{mod}\;N$ where n ≥ 2, x∈(−∞,+∞), and *N* is a large prime number, and $T_{r}{(T}_{s}{(x))\;\mathrm{mod}\;N=T}_{\mathrm{rs}}{(x)\;\mathrm{mod}\;N=T}_{s}{(T}_{r}(x))\;\mathrm{mod}\;N$.Even only with the knowledge of *x* and *y*, *n* is computationally infeasible to be obtained such that $T_{n}(x)\;\mathrm{mod}\;N=y$, which is the so-called Chaotic maps-based discrete logarithm problem (CMDLP).Even only with the knowledge of (*x*
$T_{r}(x)\;\mathrm{mod}\;N$, $T_{s}(x)\;\mathrm{mod}\;N$), $T_{\mathrm{rs}}(x)\;\mathrm{mod}\;N$ is computationally infeasible to be obtained, which is the so-called Chaotic maps-based Diffie-Hellman problem (CMDHP).

The proposed scheme applies the extended Chebyshev chaotic maps, which satisfies the above definitions.

### 3.2. System Initialization Phase

User $U_{i}$ sets up smartcard by entering an identifier and password in the system initialization phase. Server $S_{j}$ sets up the system’s parameters by performing the following steps.

Step 1.Server $S_{j}$ generates a secret value $x_{S_{j}}$, a big prime *p*, and a random number x ∈ (−∞,+∞).Step 2.Server $S_{j}$ choses a symmetric encryption algorithm $E_{k}(.)$, a symmetric decryption algorithm $D_{k}(.)$, a collision-resistance one-way hash function *H*(.), and a collision-resistance secure one-way chaotic hash function $h_{k}$(.).

### 3.3. Registration Phase

User $U_{i}$ and server $S_{j}$ perform the following steps to complete the registration phase to become a legitimate member, as illustrated in Figure 3.

Step 1.User $U_{i}$ enters ${\mathrm{ID}}_{i}$ and ${\mathrm{PW}}_{i}$.Step 2.User $U_{i}$ uses the smartcard to choose a random number $y_{i}\;\in {\;Z}_{p}^{*}.$ After that, smartcard computes ($\alpha_{i},$
$A_{i}$) as below. Finally, smartcard stores $y_{i}$ and sends (${\mathrm{ID}}_{i}$, $A_{i}$) to server $S_{j}$.
(1)$$\alpha_{i}{=T}_{y_{i}}(x)\;\mathrm{mod}\;p$$
(2)$$A_{i}{=h}_{\alpha_{i}}{(\mathrm{PW}}_{i})⊕h_{\alpha_{i}}{(y}_{i}{,\;\mathrm{SID}}_{j})$$Step 3.After receiving (${\mathrm{ID}}_{i}$, $A_{i}$), server $S_{j}$ computes elements below. Then, server $S_{j}$ returns ($B_{i}$, $B_{j}$) to user $U_{i}$.
(3)$$\beta_{j}{=T}_{x_{S_{j}}}(x)\;\mathrm{mod}\;p$$
(4)$$u_{i}{=h}_{\beta_{j}}{(\mathrm{ID}}_{i})$$
(5)$$u_{j}{=h}_{\beta_{j}}{(\mathrm{SID}}_{i})$$
(6)$$B_{i}{=U}_{i}⊕A_{i}$$
(7)$$B_{j}{=U}_{j}⊕A_{i}$$Step 4.Upon receiving ($B_{i}$, $B_{j}$), user $U_{i}$ stores ($B_{i}$, $B_{j}$) in USB or smartcard.

### 3.4. Authenticated Key Exchange Phase

To complete the mutual authentication and session key confirmation and obtain the remote server’s services, user $U_{i}$, user $U_{i}$’s smartcard, and a server $S_{j}$ perform the following steps, as illustrated in Figure 4.

Step 1.User $U_{i}$ enters ${\mathrm{ID}}_{i}$ and ${\mathrm{PW}}_{i}$.Step 2.Smartcard checks ${\mathrm{PW}}_{i}$, utilizes ($y_{i}$, *x*) to compute $A_{i}$, retrieves ($B_{i}$, $B_{j}$) to recover $u_{i}$, and computes ($K_{i}$, $R_{i}$), as below.
(8)$$u_{i}{=B}_{i}⊕A_{i}$$
(9)$$K_{i}{=A}_{i}⊕h_{\alpha_{i}}{(y}_{i})$$
(10)$$R_{i}{=B}_{j}⊕h_{\alpha_{i}}{(y}_{i})$$Step 3.Smartcard chooses integer $\rho_{i}\;\in \;(-\infty ,+\infty )$ and a big prime $N_{i}$ to compute ($\mu_{i}$, $b_{i}$, $C_{i}$) as below, and sends ($R_{i}$, $C_{i}$, $N_{i}$) to server $S_{j}$.
(11)$$\mu_{i}{=T}_{y_{i}}{(\rho }_{i}{)\;\mathrm{mod}\;N}_{i}$$
(12)$$b_{i}{=E}_{u_{i}}{(N}_{i}{||\mu }_{i})$$
(13)$$C_{i}{=E}_{K_{i}}{(\mathrm{ID}}_{i}{,\;b}_{i}{,\;\rho }_{i})$$Step 4.After receiving ($R_{i}$, $C_{i}$, $N_{i}$), server $S_{j}$ computes the equations below. If server $S_{j}$ can decrypt $b_{i}$ successfully, server $S_{j}$ successfully authenticates user $U_{i}$.
(14)$$K_{i}{=R}_{i}⊕h_{\beta_{j}}({\mathrm{SID}}_{j})$$
(15)$${(\mathrm{ID}}_{i}{,\;b}_{i}{,\;\rho }_{i}{)=D}_{K_{i}}{(C}_{i})$$
(16)$$u_{i}{=h}_{\beta_{j}}{(\mathrm{ID}}_{i})$$
(17)$${(N}_{i}{||\mu }_{i}{)=D}_{u_{i}}{(b}_{i})$$Step 5.For establishing a shared session key, server $S_{j}$ chooses a random number $s_{j}\;\in {\;Z}_{p}^{*}$, utilizes $\rho_{i}$, $N_{i}$, and $\mu_{i}$ retrieved from Step 4 to compute $\omega_{j}$, $k_{\mathrm{ji}}$, and ${\mathrm{MAC}}_{S_{j}}$, and sends $\left({\mathrm{MAC}}_{S_{j}}{,\;\omega }_{j}\right)$ to user $U_{i}$.
(18)$$\omega_{j}{=T}_{s_{j}}{(\rho }_{i}{)\;\mathrm{mod}\;N}_{i}$$
(19)$$k_{\mathrm{ji}}=H(T_{s_{j}}(\mu_{i})\;{\mathrm{mod}\;N}_{i})$$
(20)$${\mathrm{MAC}}_{S_{j}}{=h}_{k_{\mathrm{ji}}}({\mathrm{SID}}_{j}{,\;\mathrm{ID}}_{i},\;\mu_{i})$$Step 6.Upon receiving $\left({\mathrm{MAC}}_{S_{j}}{,\;\omega }_{j}\right)$, user $U_{i}$’s smartcard computes $k_{\mathrm{ij}}$ and checks whether ${\mathrm{MAC}}_{S_{j}}$ is correct. If it holds, the mutually shared session key is correct. Then, user *U_i_*’s smartcard computes ${\mathrm{MAC}}_{U_{i}}$ and sends it to server $S_{j}$.
(21)$$k_{\mathrm{ij}}=H(T_{y_{i}}{(\omega }_{j}{)\;\mathrm{mod}\;N}_{i})$$
(22)$${\mathrm{MAC}}_{S_{j}}{\;?=h}_{k_{\mathrm{ji}}}({\mathrm{SID}}_{j}{,\;\mathrm{ID}}_{i},\;\mu_{i})$$
(23)$${\mathrm{MAC}}_{U_{i}}{=h}_{k_{\mathrm{ij}}}{(\mathrm{ID}}_{i}{,\;\mathrm{SID}}_{j}{,\;\omega }_{j})$$Step 7.Upon receiving ${\mathrm{MAC}}_{U_{i}}$, server $S_{j}$ checks whether ${\mathrm{MAC}}_{U_{i}}$ is correct. If it holds, the shared session key confirmation is complete.
(24)$${\mathrm{MAC}}_{U_{i}}{\;?=h}_{k_{\mathrm{ij}}}{(\mathrm{ID}}_{i}{,\;\mathrm{SID}}_{j}{,\;\omega }_{j})$$

### 3.5. Offline Password Change Phase

User $U_{i}$ and smartcard cooperatively perform the following steps to complete the password changing process, as illustrated in Figure 5.

Step 1.User $U_{i}$ enters PIN to start smartcard and inputs old ${\mathrm{PW}}_{i}$ and new ${{\mathrm{PW}}^{\prime }}_{i}$.Step 2.Smartcard updates $A_{i}$ and stores it.
(25)$$A_{i}{=h}_{\alpha_{i}}{({\mathrm{PW}}^{\prime }}_{i})⊕h_{\alpha_{i}}{(\mathrm{PW}}_{i})⊕h_{\alpha_{i}}{(y}_{i}{,\;\mathrm{SID}}_{j})$$

## 4. Security Analysis

We apply BAN logic [52] and AVISPA tool [53] for formal security proof. We also present informal security proof, which proves that the proposed scheme can achieve some security requirements.

### 4.1. Formal Security Proof Using BAN Logic

This subsection describes the logical analyses of the proposed scheme by using the logical tool defined by Burrows et al. [52]. The process of proof in this section is similar with some schemes, because these schemes, including the proposed scheme, aim to prove that the principles in schemes can believe the established session keys. The notations used in the BAN logic [52] analysis are defined in Table 2.

#### 4.1.1. Initial Assumptions

Making initial assumptions is necessary for ensuring success of scheme and establishing the foundation of logical proof [52]. Initial assumptions of the proposed scheme are listed below.

A1. $P\in {r(C}_{P,\;Q})$: *P* can read from channel $C_{P,\;Q}$.A2. ${P\;\mathrm{believes}\;w(C}_{P,\;Q})=\{P,\;Q\}$: *P* believes that *P* and *Q* can write on $C_{P,\;Q}$.A3. P believes Q once said (Φ→Φ): *P* believes that *Q* only says what it believes.A4. ${P\;\mathrm{believes}\;\#(N}_{P})$: *P* believes that $N_{P}$ is fresh.A5. $P\;\mathrm{believes}\;({\overset{a}{\to }}_{\mathrm{ECMDH}(\mathrm{secret})}P)$: *P* believes that *a* is *P*’s extended chaotic maps-based Diffie-Hellman secret [24,49].

#### 4.1.2. Inference Rules

The purpose of inference rules is analyzing belief, which pays attention to beliefs of principals in authentication and key agreement schemes, in order to verify message, freshness, and trustworthiness of origin of scheme [52,54,55,56,57]. We apply the seeing rules, interpretation rules, freshness rules, and the rationality rules for logical proof.

The seeing rules define that if a principle sees a formula, the principle also sees its components with knowing necessary keys. We apply S1 and S2 as below.

S1. $\frac{P\;\mathrm{sees}\;C(X),\;P\in r(C)}{P\;\mathrm{believes}\;(P\;\mathrm{sees}\;X|C),\;P\;\mathrm{sees}\;X}$: If *P* receives and reads *X* via *C*, then *P* believes that *X* has arrived on *C* and *P* sees *X*.S2. $\frac{P\;\mathrm{sees}\;C(X,\;Y)}{P\;\mathrm{sees}\;X,\;P\;\mathrm{sees}\;Y}$: If *P* sees a hybrid message (*X*, *Y*), then *P* sees *X* and *Y* separately.

The interpretation rules define that a principle can believe some hybrid facts by logical reasoning. We apply I1, I2, and I3, as below.

I1. $\frac{P\;\mathrm{believes}\;(w(C)=\{P,\;Q\})}{P\;\mathrm{believes}\;(P\;\mathrm{sees}\;X|C)\to Q\;\mathrm{once}\;\mathrm{said}\;X}$: If *P* believes that *C* can only be written by *P* and *Q*, then *P* believes that if *P* receives *X* via *C*, then *Q* said *X*.I2. $\frac{P\;\mathrm{believes}\;(Q\;\mathrm{once}\;\mathrm{said}\;(X,\;Y))}{P\;\mathrm{believes}\;(Q\;\mathrm{once}\;\mathrm{said}\;X),\;P\;\mathrm{believes}\;(Q\;\mathrm{once}\;\mathrm{said}\;Y)}$: If *P* believes that *Q* said a hybrid message (*X*, *Y*), then *P* believes that *Q* has said *X* and *Y* separately.I3. $\frac{P\;\mathrm{believes}\;({\overset{a}{\to }}_{\mathrm{ECMDH}(\mathrm{secret})}P),\;P\;\mathrm{believes}\;({\overset{T_{b}(x)\;\mathrm{mod}\;N}{\to }}_{\mathrm{ECMDH}(\mathrm{public})}Q)}{P\;\mathrm{believes}\;(P\overset{T_{\mathrm{ab}}(x)\;\mathrm{mod}\;N}{↔}Q)}$: If *P* believes that *a* is *P*’s extended chaotic maps-based Diffie-Hellman secret and $T_{b}(x)\;\mathrm{mod}\;N$ is extended chaotic maps-based Diffie-Hellman component from *Q*, then *P* believes that $T_{\mathrm{ab}}(x)\;\mathrm{mod}\;N$ is symmetric key shared between *P* and *Q*.

The freshness rules define that if one part of a formula is fresh, the entire formula must be fresh. We apply F1 and F2 as below.

F1. $\frac{P\;\mathrm{believes}\;(Q\;\mathrm{once}\;\mathrm{said}\;X),\;P\;\mathrm{believes}\;\#(X)}{P\;\mathrm{believes}\;(Q\;\mathrm{once}\;\mathrm{said}\;X)}$: If *P* believes that another *Q* said *X* and *P* also believes that *X* is fresh, then *P* believes that *Q* recently said *X*.F2. $\frac{P\;\mathrm{believes}\;\#(X)}{P\;\mathrm{believes}\;\#(X,\;Y)}$: If *P* believes that a part of a mixed message *X* is fresh, then it believes that the whole message (*X*, *Y*) is fresh.

The rationality rules define that a principle can only believe what it believes. We have R1 as below.

R1. $\frac{{P\;\mathrm{believes}\;(\Phi }_{1}\to \Phi_{2}{),\;P\;\mathrm{believes}\;\Phi }_{1}}{{P\;\mathrm{believes}\;\Phi }_{2}}$: If *P* believes that $\Phi_{1}$ implies $\Phi_{2}$ and *P* believes that $\Phi_{1}$ is true, then *P* believes that $\Phi_{2}$ is true.

#### 4.1.3. Goals

Goals are what schemes must achieve, and goals are required while designing schemes. The goals of the proposed scheme are listed below.

Goal 1. $U_{i}\;\mathrm{believes}\;(U_{i}\overset{T_{y_{i}s_{j}}{(\rho }_{i}{)\;\mathrm{mod}\;N}_{i}}{↔}S_{j})$: User $U_{i}$ believes that $T_{y_{i}s_{j}}(\rho_{i})\;{\mathrm{mod}\;N}_{i}$ is a symmetric key shared between participants $U_{i}$ and $S_{j}$.Goal 2. $S_{j}\;\mathrm{believes}\;(U_{i}\overset{T_{y_{i}s_{j}}{(\rho }_{i}{)\;\mathrm{mod}\;N}_{i}}{↔}S_{j})$: Server $S_{j}$ believes that $T_{y_{i}s_{j}}{(\rho }_{i}{)\;\mathrm{mod}\;N}_{i}$ is a symmetric key shared between participants $U_{i}$ and $S_{j}$.Goal 3. $U_{i}{\;\mathrm{believes}\;S}_{j}\;\mathrm{believes}\;(U_{i}\overset{T_{y_{i}s_{j}}{(\rho }_{i}{)\;\mathrm{mod}\;N}_{i}}{↔}S_{j})$: User $U_{i}$ believes that $S_{j}$ believes $T_{y_{i}s_{j}}(\rho_{i})\;{\mathrm{mod}\;N}_{i}$ is a symmetric key shared between $U_{i}$ and $S_{j}$.Goal 4. $S_{j}{\;\mathrm{believes}\;U}_{i}\;\mathrm{believes}\;(U_{i}\overset{T_{y_{i}s_{j}}{(\rho }_{i}{)\;\mathrm{mod}\;N}_{i}}{↔}S_{j})$: Server $S_{j}$ believes that $U_{i}$ believes $T_{y_{i}s_{j}}{(\rho }_{i}{)\;\mathrm{mod}\;N}_{i}$ is a symmetric key shared between $U_{i}$ and $S_{j}$.

#### 4.1.4. Proof

The proposed scheme can be normalized as Steps 1 and 2.

Step 1.
$S_{j}\;\mathrm{sees}\;({\overset{T_{y_{i}}(\rho_{i}){\;\mathrm{mod}\;N}_{i}}{\to }}_{\mathrm{ECMDH}(\mathrm{public})}U_{i}{,C}_{S_{j}{,\;U}_{i}}({\mathrm{ID}}_{i}{||b}_{i}{||x}_{i2})K_{i}{,\;N}_{i})$
Step 2.
$U_{i}\;\mathrm{sees}\;({\overset{T_{s_{j}}{(\rho }_{i}{)\;\mathrm{mod}\;N}_{i}}{\to }}_{\mathrm{ECMDH}(\mathrm{public})}S_{j}{,C}_{U_{i}{,\;S}_{j}}{\left({\mathrm{SID}}_{j}{,\;\mathrm{ID}}_{i}{,\;T}_{y_{i}}{(\rho }_{i}{)\;\mathrm{mod}\;N}_{i}\right)}_{k_{\mathrm{ij}}}{,\;T}_{s_{j}}{(\rho }_{i}{)\;\mathrm{mod}\;N}_{i})$


Equation (26) means user $U_{i}$ believes that $y_{i}$ is its extended chaotic maps-based Diffie-Hellman secret. Equation (27) means user $U_{i}$ believes that $T_{s_{j}}{(\rho }_{i}{)\;\mathrm{mod}\;N}_{i}$ is the extended chaotic maps-based Diffie-Hellman component from server $S_{j}$*_._* To accomplish Goal 1 (User $U_{i}$ believes that $k_{\mathrm{ij}}{=T}_{y_{i}s_{j}}{(\rho }_{i}{)\;\mathrm{mod}\;N}_{i}$ is a symmetric key shared between participants user $U_{i}$ and server $S_{j}$), Equations (25) and (26) must hold, because of the interpretation rule (I3) and assumption (A5).
(26)$$U_{i}\;\mathrm{believes}\;({\overset{y_{i}}{\to }}_{\mathrm{ECDHM}(\mathrm{secret})}U_{i})$$
(27)$$U_{i}\;\mathrm{believes}\;({\overset{T_{s_{j}}{(\rho }_{i}{)\;\mathrm{mod}\;N}_{i}}{\to }}_{\mathrm{ECDHM}(\mathrm{public})}S_{j})$$

The meaning of Equation (28) is described below. The first fact is that server $S_{j}$ once said that $T_{s_{j}}$(*x*) mod *p* is the extended chaotic maps-based Diffie-Hellman public component from server $S_{j}$, $\left({\mathrm{SID}}_{j}{,\;\mathrm{ID}}_{i}{,\;T}_{y_{i}}{(\rho }_{i}{)\;\mathrm{mod}\;N}_{i}\right)$ is encrypted by $k_{\mathrm{ij}}$ and $T_{s_{j}}{(\rho i)\;\mathrm{mod}\;N}_{i}$. The second fact is that server $S_{j}$ once said that $T_{s_{j}}$(*x*) mod *p* is the extended chaotic maps-based Diffie-Hellman public component from server $S_{j}$. In Equation (29), user $U_{i}$ believes that the first fact implies the second fact. Equation (28) means that user $U_{i}$ believes that server $S_{j}$ once said that $T_{s_{j}}{(\rho }_{i}{)\;\mathrm{mod}\;N}_{i}$ is the extended chaotic maps-based Diffie-Hellman public component from server $S_{j}$. Next, to accomplish Equation (27), Equations (28) and (29) must hold because of assumption (A3) and the rationality rule (R1).
(28)$$U_{i}\;\mathrm{believes}\;(S_{j}\;\mathrm{once}\;\mathrm{said}\;({\overset{T_{s_{j}}{(\rho }_{i}{)\;\mathrm{mod}\;N}_{i}}{\to }}_{\mathrm{ECDHM}(\mathrm{public})}S_{j}{,\;(\mathrm{SID}}_{j}{,\;\mathrm{ID}}_{i}{,\;T}_{y_{i}}{(\rho }_{i}{)\;\mathrm{mod}\;N}_{i})k_{\mathrm{ij}}{,\;T}_{s_{j}}{(\rho }_{i})\;\mathrm{mod}\;p)\to ({\overset{T_{s_{j}}{(\rho }_{i}{)\;\mathrm{mod}\;N}_{i}}{\to }}_{\mathrm{ECDHM}(\mathrm{public})}S_{j}))$$
(29)$$U_{i}\;\mathrm{believes}\;(S_{j}\;\mathrm{once}\;\mathrm{said}\;({\overset{T_{s_{j}}{(\rho }_{i}{)\;\mathrm{mod}\;N}_{i}}{\to }}_{\mathrm{ECDHM}(\mathrm{public})}S_{j}))$$

To accomplish Equation (29), Equation (30) must hold, which means that user $U_{i}$ believes that $T_{s_{j}}{(\rho }_{i}{)\;\mathrm{mod}\;N}_{i}$, which is that the extended chaotic maps-based Diffie-Hellman public component from server $S_{j}$ is fresh because of freshness rules (F1) and (F2), and assumption (A4).
(30)$$U_{i}\;\mathrm{believes}\;\#({\overset{T_{s_{j}}{(\rho }_{i}{)\;\mathrm{mod}\;N}_{i}}{\to }}_{\mathrm{ECDHM}(\mathrm{public})}S_{j})$$

Equation (31) means that user $U_{i}$ can read from the channel $C_{S_{j}{,\;U}_{i}}$. Equation (32) means that user $U_{i}$ believes that user $U_{i}$ and server $S_{j}$ can write messages on channel $C_{S_{j}{,\;U}_{i}}$. Equation (33) means that user *U_i_* sees and believes that $T_{s_{j}}{(\rho }_{i}{)\;\mathrm{mod}\;N}_{i}$ is in the channel $C_{S_{j}{,\;U}_{i}}$, which is the extended chaotic maps-based Diffie-Hellman public component from server $S_{j}$. To accomplish Equation (30), we have Equations (31)–(33) that must hold because of the interpretation rules (I1), the seeing rules (S1), (S2), assumptions (A1) and (A2). By using the interpretation rules (I3), our proposed scheme realizes that Goal 1 is achieved. Similarly, we ensured that the proposed scheme realizes Goal 2 by using the same arguments of Goal 1.
(31)$$U_{i}\in \;r(C_{S_{j}{,\;U}_{i}})$$
(32)$$U_{i}{\;\mathrm{believes}\;(w(C}_{S_{j}{,\;U}_{i}}{)=\{U}_{i}{,\;S}_{j}\})$$
(33)$$U_{i}{\;\mathrm{sees}\;\mathrm{believes}\;C}_{S_{j}{,\;U}_{i}}({\overset{T_{s_{j}}{(\rho }_{i}{)\;\mathrm{mod}\;N}_{i}}{\to }}_{\mathrm{ECDHM}(\mathrm{public})}S_{j})$$

The meaning of Equation (34) is described below. The first fact is that server $S_{j}$ once said that $T_{y_{i}s_{j}}{(\rho }_{i}{)\;\mathrm{mod}\;N}_{i}$ is the symmetric key shared between $U_{i}$ and $S_{j}$. The second fact is that server $S_{j}$ believes that $T_{y_{i}s_{j}}{(\rho }_{i}{)\;\mathrm{mod}\;N}_{i}$ is the symmetric key shared between $U_{i}$ and $S_{j}$. In Equation (35), user $U_{i}$ believes that the first fact implies the second fact. To accomplish the Goal 3, we have Equations (34) and (35), which must hold because of the rationality rule (R1) and assumption (A3).
(34)$$U_{i}{\;\mathrm{believes}\;((S}_{j}{\;\mathrm{once}\;\mathrm{said}\;U}_{i}\overset{T_{y_{i}s_{j}}{(\rho }_{i}{)\;\mathrm{mod}\;N}_{i}}{↔}S_{j})\to S_{j}{\;\mathrm{believes}\;(U}_{i}\overset{T_{y_{i}s_{j}}{(\rho }_{i}{)\;\mathrm{mod}\;N}_{i}}{↔}S_{j}))$$
(35)$$U_{i}{\;\mathrm{believes}\;(S}_{j}{\;\mathrm{once}\;\mathrm{said}\;U}_{i}\overset{T_{y_{i}s_{j}}{(\rho }_{i}{)\;\mathrm{mod}\;N}_{i}}{↔}S_{j})$$

Equation (36) means that user $U_{i}$ believes symmetric key $T_{y_{i}s_{j}}{(\rho }_{i}{)\;\mathrm{mod}\;N}_{i}$ is fresh. To accomplish Equation (35), Equation (36) must hold because of the freshness rules (F1) and (F2) and assumption (A4).
(36)$$U_{i}{\;\mathrm{believes}\;\#(U}_{i}\overset{T_{y_{i}s_{j}}{(\rho }_{i}{)\;\mathrm{mod}\;N}_{i}}{↔}S_{j})$$

Equation (37) means that user $U_{i}$ sees and believes that $T_{y_{i}s_{j}}{(\rho }_{i}{)\;\mathrm{mod}\;N}_{i}$ is in the channel $C_{S_{j}{,\;U}_{i}}$. To accomplish Equation (36), Equations (31), (32) and (37) must hold because of the interpretation rule (I1), the assumptions (A1) and (A2), and the seeing rules (S1) and (S2). Thus, the proposed scheme realizes that Goal 3 is achieved. Similarly, using the same arguments of Goal 3, the proposed scheme realizes Goal 4.
(37)$$U_{i}{\;\mathrm{sees}\;\mathrm{believes}\;C}_{S_{j}{,\;U}_{i}}{(U}_{i}\overset{T_{y_{i}s_{j}}{(\rho }_{i}{)\;\mathrm{mod}\;N}_{i}}{↔}S_{j})$$

Therefore, the proposed scheme realizes Goals 1, 2, 3, and 4.

### 4.2. Formal Security Proof Using AVISPA

Automated validation of internet security protocols and applications (AVISPA) is a high-level language tool for security protocols, and it provides automatic analysis techniques through its back-ends, called on-the-fly model-checker (OFMC), constraint logic based attack searcher (CL-AtSe), SAT-based model-checker (SATMC), and tree automata based on automatic approximations for the analysis of security protocols (TA4SP) [53,58,59,60]. The AVISPA tool executes a simulated protocol through high-level protocol specification language (HLPSL) [61]. We used the AVISPA tool to verify the proposed scheme. The HLPSL specification of user U and server S are shown in Figure 6 and Figure 7, respectively. The session role, environment role, and goals are also specified in HLPSL, shown in Figure 8. Figure 9 shows the results and proves that the proposed scheme is safe.

### 4.3. Informal Security Proof

We present theoretical analyses that proved that proposed scheme could achieve security requirements.

#### 4.3.1. Preventing MITM Attack

In order to prevent MITM attack, user $U_{i}$ and server $S_{j}$ can confirm whether the message is resent, modified, and replaced, by checking information through message authentication codes ${\mathrm{MAC}}_{S_{j}}$ and ${\mathrm{MAC}}_{U_{i}}$. User $U_{i}$ verifies ${\mathrm{MAC}}_{S_{j}}{=h}_{k_{\mathrm{ji}}}{(\mathrm{SID}}_{j}{,\;\mathrm{ID}}_{i},\;\mu_{i})$ at Step 6, and server $S_{j}$ verifies ${\mathrm{MAC}}_{U_{i}}{=h}_{k_{\mathrm{ij}}}{(\mathrm{ID}}_{i}{,\;\mathrm{SID}}_{j}{,\;\omega }_{j})$ at Step 7 in the authenticated key exchange phase of the proposed scheme. In this way, the adversary cannot modify message authentication codes ${\mathrm{MAC}}_{S_{j}}$ and ${\mathrm{MAC}}_{U_{i}}$ without session key $k_{\mathrm{ij}}$.Thus, the proposed scheme can prevent MITM attack.

#### 4.3.2. Key Confirmation

User $U_{i}$ can check session key $k_{\mathrm{ij}}$ by ${\mathrm{MAC}}_{S_{j}}{\;?=h}_{k_{\mathrm{ji}}}{(\mathrm{SID}}_{j}{,\;\mathrm{ID}}_{i},\;\mu_{i})$, and server $S_{j}$ can also check session key $k_{\mathrm{ji}}$ through ${\mathrm{MAC}}_{U_{i}}{\;?=h}_{k_{\mathrm{ij}}}{(\mathrm{ID}}_{i}{,\;\mathrm{SID}}_{j}{,\;\omega }_{j})$ in the proposed scheme. As a result, the proposed scheme achieves key confirmation.

#### 4.3.3. Preventing Key-Compromise Impersonation and Server Spoofing Attacks

User $U_{i}$’s random number $y_{i}$ is stored in a smartcard, which is hard to obtain information. The adversary must have user $U_{i}$’s smartcard and correct password if they want to impersonate a legitimate user. The number of attempts that a password can be entered is limited; if the number of attempts to enter a password exceeds the allowable number of attempts, the smartcard will get locked. On the other hand, the adversary cannot obtain $K_{i}$ due to not knowing $x_{S_{j}}$, and afterwards the process cannot be completed by adversary. As a result, the proposed scheme can prevent key-compromise impersonation and server spoofing attacks.

#### 4.3.4. Mutual Authentication

In the authenticated key exchange phase of the proposed scheme, server $S_{j}$ encrypts ${(\mathrm{SID}}_{j}{,\;\mathrm{ID}}_{i},\;\mu_{i})$ from user $U_{i}$ to message authentication code ${\mathrm{MAC}}_{S_{j}}$ with session key $k_{\mathrm{ji}}{=H(T}_{s_{j}}(\mu_{i})\;\mathrm{mod}\;\mathrm{Ni})$ and sends ${(\mathrm{MAC}}_{S_{j}}{,\;\omega }_{j})$ to user $U_{i}$. In Step 6, user $U_{i}$ uses $\omega_{j}$ from server $S_{j}$ to obtain session key $k_{\mathrm{ij}}$ and verify ${\mathrm{MAC}}_{S_{j}}{=h}_{k_{\mathrm{ji}}}{(\mathrm{SID}}_{j}{,\;\mathrm{ID}}_{i}{,\;\omega }_{j})$. Server $S_{j}$ verifies message authentication code ${\mathrm{MAC}}_{U_{i}}{=h}_{k_{\mathrm{ij}}}{(\mathrm{ID}}_{i}{,\;\mathrm{SID}}_{j}{,\;\omega }_{j})$ sent by user $U_{i}$ in Step 7. ${\mathrm{MAC}}_{S_{j}}$ and ${\mathrm{MAC}}_{U_{i}}$ are included in session keys that only two parties of communication have, so only user $U_{i}$ and server $S_{j}$ can verify each other.

#### 4.3.5. User Anonymity

User $U_{i}$’s identity ${\mathrm{ID}}_{i}$ is protected by being encrypted in $C_{i}{=E}_{K_{i}}{(\mathrm{ID}}_{i}{,\;b}_{i}{,\;\rho }_{i})$ with $K_{i}$, before being sent. Server $S_{j}$ must obtain $K_{i}$ by computing $K_{i}{=R}_{i}⊕h_{\beta_{j}}{(\mathrm{SID}}_{j})$. The adversary cannot obtain ${\mathrm{ID}}_{i}$ even with $R_{i}$ and $C_{i}$ because only server $S_{j}$ has knowledge of secret $x_{S_{j}}$. The adversary cannot obtain $K_{i}$ without $x_{S_{j}}$ and decrypting $C_{i}$; thus, the adversary cannot obtain ${\mathrm{ID}}_{i}$. As a result, the proposed scheme provides user anonymity during communication.

#### 4.3.6. Resistant to Bergamo et al.’s Attack

Bergamo et al.’s attack is based on [62]. (i) The adversary is able to obtain related elements (*x*, $\rho_{i}$, $\mu_{i}$, $\omega_{j}$); and (ii) several Chebyshev polynomials pass through the same point due to periodicity of the cosine function. In the proposed scheme, the adversary is unable to obtain any related elements (*x*, $\rho_{i}$, $\mu_{i}$, $\omega_{j}$) as these are encrypted in transmitted messages where only user $U_{i}$ and server $S_{j}$ can retrieve decryption key. Moreover, the proposed protocol utilizes the extended Chebyshev polynomials, in which the periodicity of the cosine function is avoided by extending the interval of *x* to (−∞,+∞) [51]. As a result, the proposed scheme can resist the attack proposed by Bergamo et al. [62].

## 5. Performance Analysis

We present relevant security requirements and computational complexity comparison.

### 5.1. Comparisons of Security Requirements

Table 3 shows comparisons of security requirements that were presented in the schemes designed by Wang and Zhao [63], Yoon and Jeon [46], Lin [48], Lin and Zhu [64], Lee et al. [49], Madhusudhan et al. [65], Sureshkumar et al. [38], and us. Wang and Zhao’s [63], Lin’s [48], and Lin and Zhu’s schemes [64] are not secure against key-compromise impersonation attack, since the transmitted messages can be replayed by an adversary. Wang and Zhao’s [63], Madhusudhan et al.’s [65], and Sureshkumar et al.’s [38] scheme cannot prevent server spoofing attack. Our scheme is secure against both key-compromise impersonation attack and server spoofing attack. Furthermore, our scheme provides user anonymity, which Wang and Zhao’s [63], Yoon and Jeon’s [46], and Madhusudhan et al.’s [65] scheme do not. Our scheme can also prevent MITM attack, which Wang and Zhao’s [63], Yoon and Jeon’s [46], and Madhusudhan et al.’s [65] scheme cannot. Furthermore, our scheme ensures that users and servers use the same shared key in a session via key confirmation, which is not present in Wang and Zhao’s [63], Yoon and Jeon’s [46], Lin’s [48], Lin and Zhu’s [64], Lee et al.’s [49], Madhusudhan et al.’s [65], and Sureshkumar et al.’s [38] scheme. Moreover, our scheme can prevent DoS attacks, which Wang and Zhao’s [63], Yoon and Jeon’s [46], Lin’s [48], and Madhusudhan et al.’s [65] scheme cannot.

### 5.2. Comparisons of Computational Complexity

We present the computational complexity comparison with Lee et al.’s [49], Madhusudhan et al.’s [65], and Sureshkumar et al.’s [38] scheme, as shown in Table 4. We ignore the time taken for computing XOR operations because the value is too low to influence result. Although our scheme needs more one-way hash function operations than Lee et al.’s [49] scheme and more symmetry encryption operations than Lee et al.’s [49], Madhusudhan et al.’s [65], and Sureshkumar et al.’s [38] scheme, our scheme allows key confirmation. Even so, our scheme has the less overall computational cost than Lee et al.’s [49], Madhusudhan et al.’s [65], and Sureshkumar et al.’s [38] scheme. Users in our scheme can enjoy telemedicine services witha lower computational cost. As a result, our scheme is more efficient than Lee et al.’s [49], Madhusudhan et al.’s [65], and Sureshkumar et al.’s [38]. Figure 10 illustrates the computational complexity of the server with varying number of users, and Figure 11 illustrates the computational complexity of user with varying number of servers. The computational complexity of user in Lee et al.’s scheme [49] is related to the number of servers. Computational complexity of user in Madhusudhan et al.’s [65], Sureshkumar et al.’s [38], and the proposed scheme is not related to number of servers, and the proposed scheme shows the least computational complexity among the compared schemes.

## 6. Implementation

We developed SC-UCSSO system using the proposed scheme, in a multi-function smart token, as shown in Figure 12, which supports the public key infrastructure and the X.509 certificate. A user can insert a smart token to a computer or a laptop and insert the smartcard shown in Figure 13 into a smart token, in order to use the system. Figure 14 and Figure 15 show the registration and login interfaces. Figure 16 shows that the user can login to multiple services, which implies that the proposed system can be used in multi-server environments. The proposed system also provides account checking (Figure 17) to manage the user’s accounts. The user can login to the online telemedicine website using a computer, laptop, smartphone, or any wireless devices that has a webcam with a smart token and a smartcard in synchronous telemedicine scenario. The channel of online video consult between the patient and medical professional is protected by the session key generated by the proposed scheme. In asynchronous telemedicine, the measured biodata is transmitted to a smartphone using Bluetooth, and the user can decide when to send data to the designed server of telemedicine systems. The user logins with smart token and smartcard, before sending data. Transmitted measured data between smartphone and servers would be protected by the session key generated through the proposed scheme. The user has data ownership because the user can control data’s destination and the time of being transmitted. Once data are sent by user, the privacy of user would be protected because the transmission channel is secure with the session key.

## 7. Discussion

We give a discussion for brief review, real-life scenario, and limitations of this research.

Telemedicine systems work in public networks, where privacy preservation issue of users and sensitive and private transmitted information is important [1]. Security issues related to data transmission are discussed, such as eavesdropping, MITM attack, data tempering attack, message modification attack, data interception attack, etc. [12] Although regulations, such as HIPAA, GDPR, Safe Harbor Laws, etc., were developed, technical support is still not enough [12,13,14]. We proposed an SC-UCSSO for the 5G-IoT telemedicine systems, which can be applied in the 5G-IoT telemedicine multi-server environments. Security of the proposed scheme was proved by BAN logic, AVISPA tool, and theoretical analyses. The proposed scheme achieved general security requirements, such as preventing MITM attack, preventing key-compromise impersonation, and server spoofing attacks, and user anonymity, key confirmation, and mutual authentication. Moreover, the proposed scheme overcomes the drawbacks of the compared previous schemes, such as stolen-verification table attack, clock synchronization problem, and DoS attack, as shown in Table 3 in the previous section. The proposed scheme applies the extended Chebychev chaotic maps that can resist Bergamo et al.’s attack [62]. Performance of the proposed scheme is also compared with Lee et al.’s [49], Madhusudhan et al.’s [65], and Sureshkumar et al.’s [38] scheme by analyzing the computational complexity of each scheme, and the results showed that the proposed scheme was less expensive (719$T_{h}$) in total than Lee et al.’s [49] (${1061T}_{h}$), Madhusudhan et al.’s [65] (1588$T_{h}$), and Sureshkumar et al.’s [38] scheme (1235$T_{h}$), as shown in Table 4.

We give four possible real-life scenarios of telemedicine systems in 5G-IoT environments that can apply the proposed scheme.

Scenario 1:Patient inserts smartcard (e.g., health insurance card or smartcard, as in Figure 13) into measurement devices that include a smartcard reader, such as sphygmomanometer or blood-glucose meter, before measuring biodata. Once a patient inserts smartcard, the authenticated key agreement phase of the proposed scheme is activated, and measured biodata can be transmitted securely to server as it is encrypted by the session key.Scenario 2:Patient’s wearable healthcare device (e.g., sensors, smart watch, etc.) transmits the measured biodata to the related mobile application (APP) on a smartphone, through data synchronization via Bluetooth, NFC, RFID, etc. If the patient wants to transmit the measured biodata to server, the patient can use a smartphone with a smartcard adopter, such as the smart token in Figure 13. Once a patient inserts the smartcard, the authenticated key agreement phase of proposed scheme is activated, and the measured biodata can be securely transmitted to server as it is encrypted by the session key.Scenario 3:Patient’s measured biodata are recorded or stored in storage at home. If the patient wants to transmit the measured biodata to server, the patient can use the smartcard with a reader. Once a patient inserts the smartcard, the authenticated key agreement phase of proposed scheme is activated, and the measured biodata can be transmitted securely to server as it is encrypted by the session key.Scenario 4:If a medical professional would like to access the measured biodata on server, the medical professional has to use the smartcard (e.g., healthcare certification IC card [66]) with a reader. Once a medical professional inserts smartcard, the authenticated key agreement phase of proposed scheme is activated, and the measured biodata can be securely transmitted as it is encrypted by the session key.

Scenario 1 to 3 allow the patient to decide the data’s destination and time of transmission.

This research has limitations. We only give a software security analysis, but hardware security and availability are other aspects of security in telemedicine systems, such as electromagnetic interference (EMI), which might affect the functions on wearable devices. Although there are already measurement devices with a smartcard reader on the market, we did not evaluate the hardware’s effects with the proposed scheme. We assumed that the users (patient/medical professional) have a smartcard (health insurance card/ healthcare certification IC card) and proposed a smartcard-based scheme, but authentication could be achieved in many ways, such as three-factor authentication, two-step verification, fast identity online (FIDO), etc., which can be related to works in the future.

## 8. Conclusions

Telemedicine systems is a multi-functional remote medical service that can help patients in bed in long-distance communication environments [1,2,3,4]. As telemedicine systems work in public networks, privacy preservation issue of sensitive and private transmitted information is important. [1]. We proposed a SC-UCSSO for 5G-IoT telemedicine systems, which could achieve some general security requirements, such as preventing MITM attack, preventing key-compromise impersonation and server spoofing attacks, provide user anonymity, and overcomes the drawbacks of the previous schemes compared herein. The proposed scheme establishes a secure communication channel using the authenticated session keys between patients and services of telemedicine systems, without threats of eavesdrop, impersonation, etc., and allow patient access to multiple telemedicine services, with a pair of identity and password. Formal security analysis using BAN logic [52] and the AVISPA tool [67] was given. We also gave a performance analysis and proved that the proposed scheme is more efficient than previous compared schemes, and computational complexity of the user in proposed scheme was not related to the number of servers. Moreover, the proposed scheme is suitable for asynchronous and synchronous telemedicine, and patients have data ownership because the user can control and decide data’s destination and time of transmission.

## Figures and Tables

**Figure 1 sensors-21-02880-f001:**
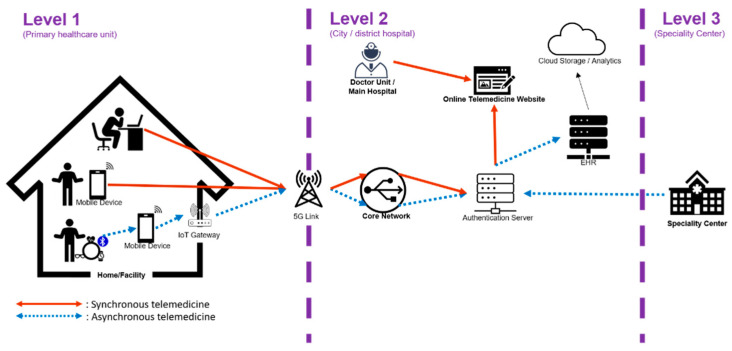
A general telemedicine system with asynchronous and synchronous telemedicine.

**Figure 2 sensors-21-02880-f002:**
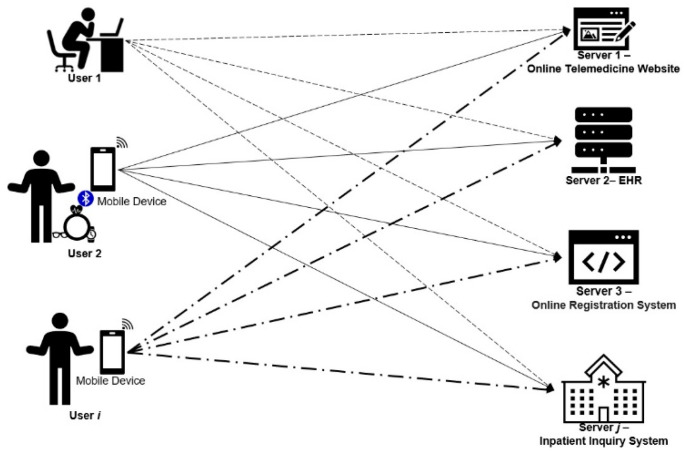
System structure of the proposed scheme.

**Figure 3 sensors-21-02880-f003:**
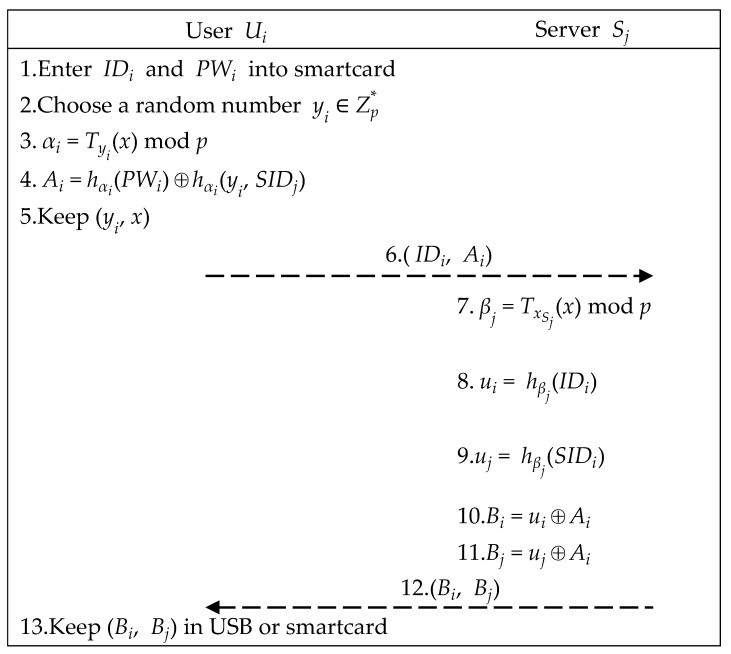
Registration phase of the proposed scheme.

**Figure 4 sensors-21-02880-f004:**
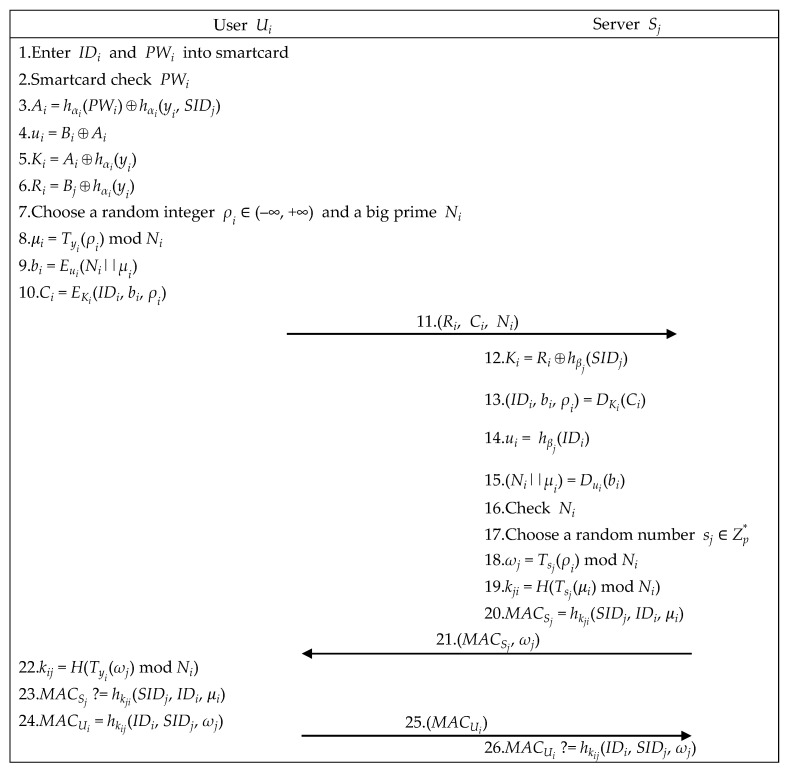
The authenticated key exchange phase of the proposed scheme.

**Figure 5 sensors-21-02880-f005:**
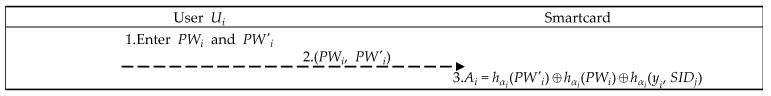
Offline password change phase of the proposed scheme.

**Figure 6 sensors-21-02880-f006:**
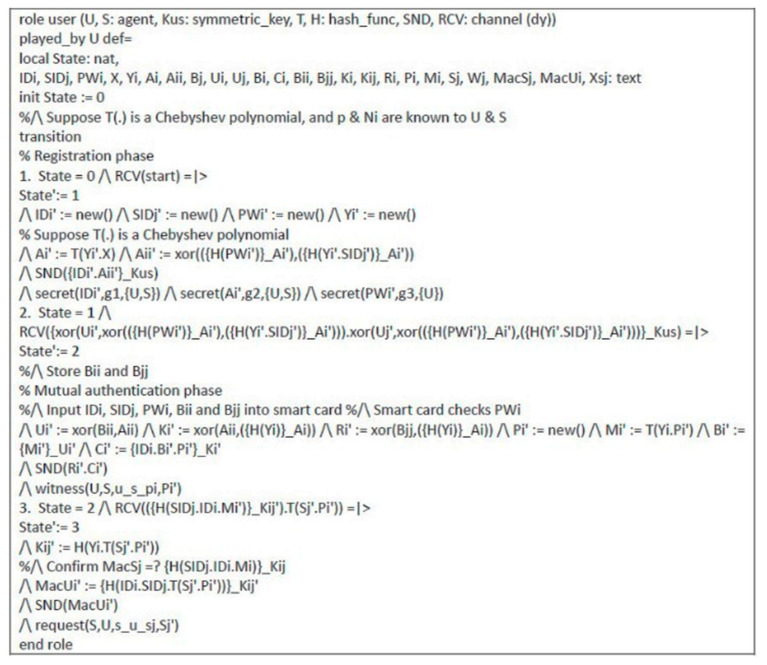
HLPSL specification of user.

**Figure 7 sensors-21-02880-f007:**
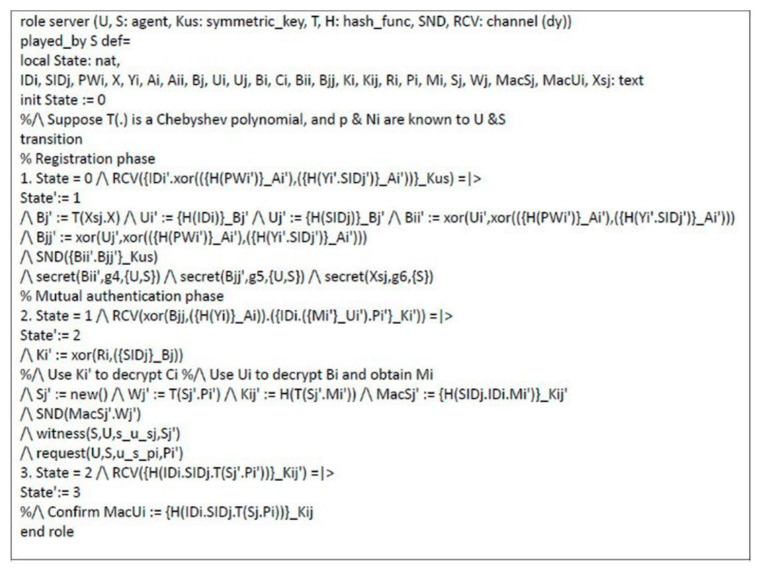
HLPSL specification of server.

**Figure 8 sensors-21-02880-f008:**
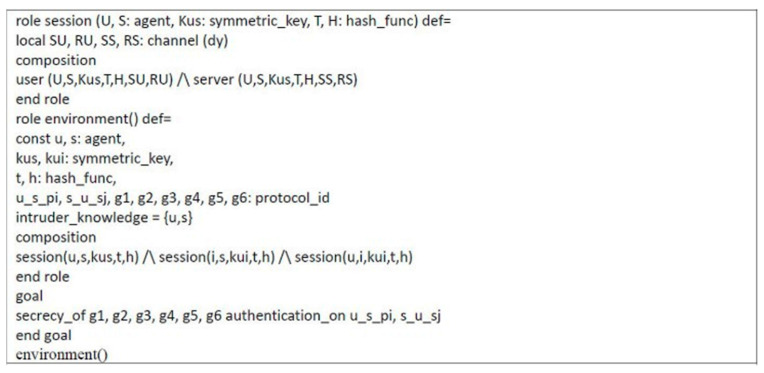
HLPSL specification of session role, environment role, and goals.

**Figure 9 sensors-21-02880-f009:**
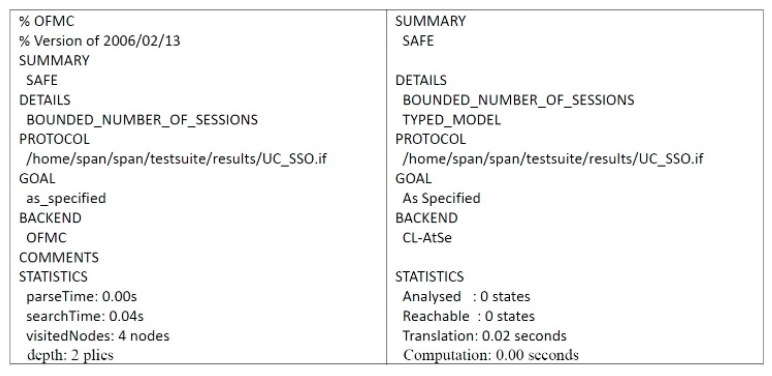
Results of AVISPA.

**Figure 10 sensors-21-02880-f010:**
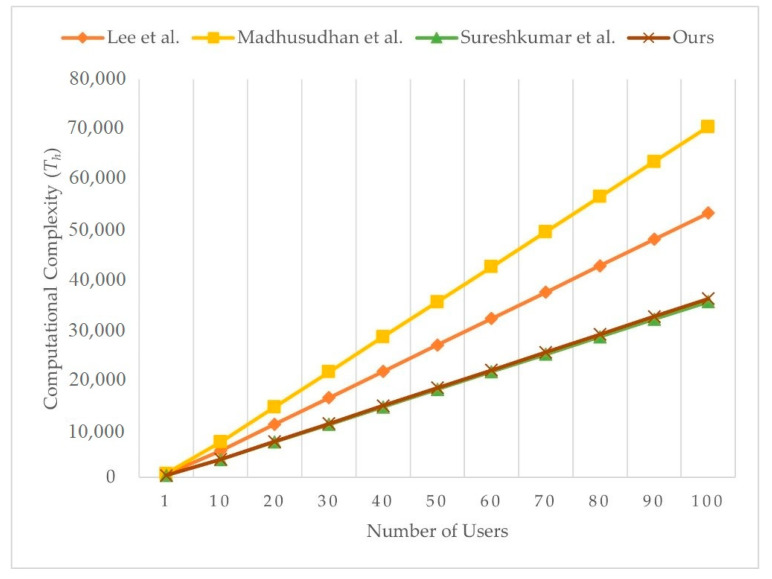
Computational complexity of server with varying number of users.

**Figure 11 sensors-21-02880-f011:**
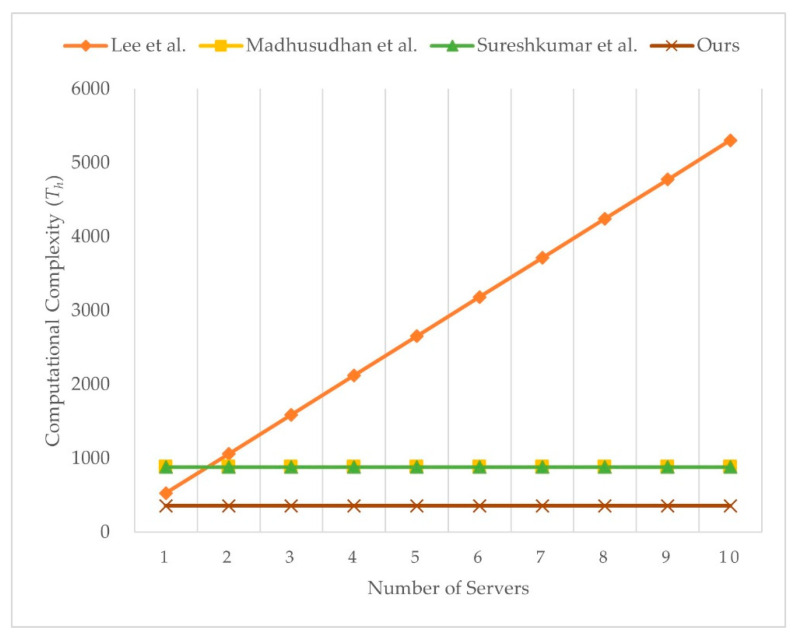
Computational complexity of user with varying number of servers.

**Figure 12 sensors-21-02880-f012:**
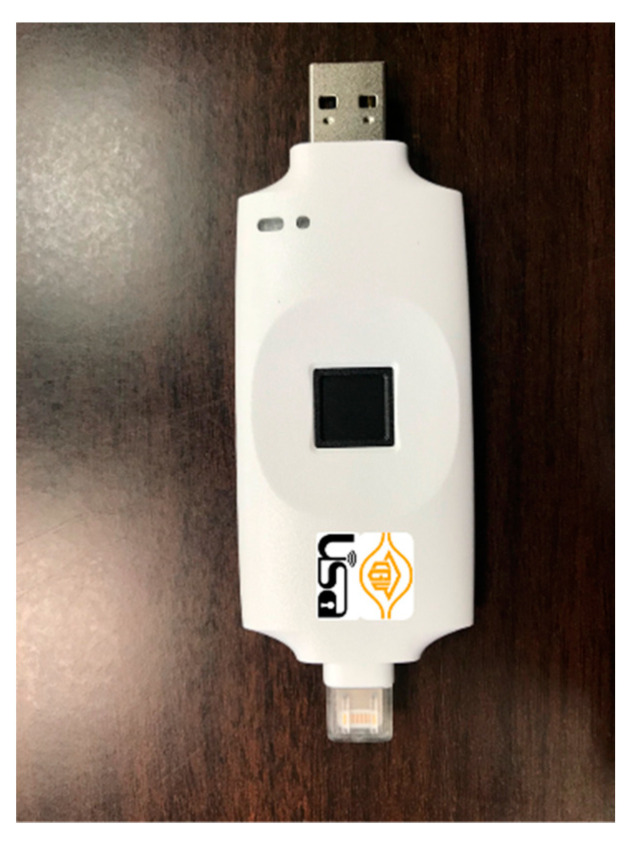
Multi-function smart token.

**Figure 13 sensors-21-02880-f013:**
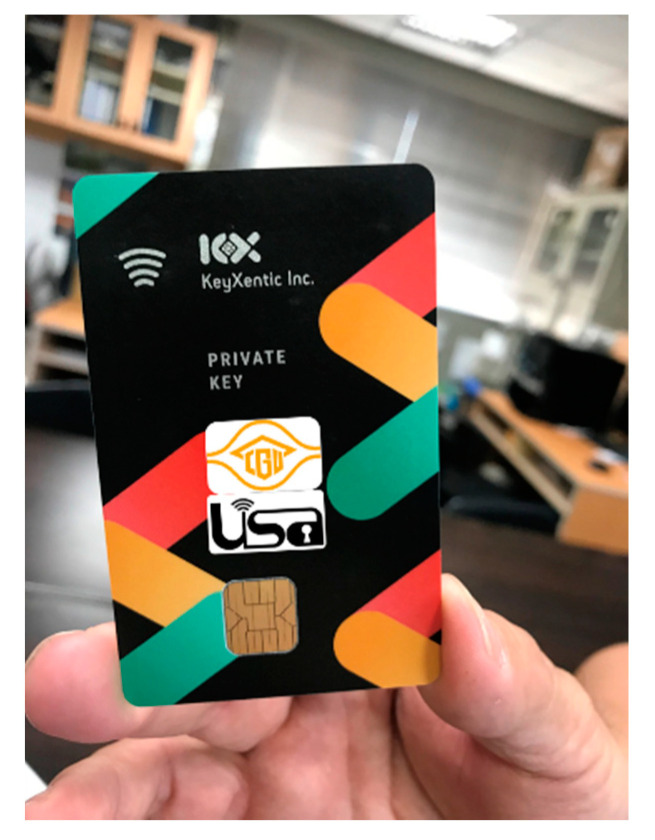
Smartcard.

**Figure 14 sensors-21-02880-f014:**
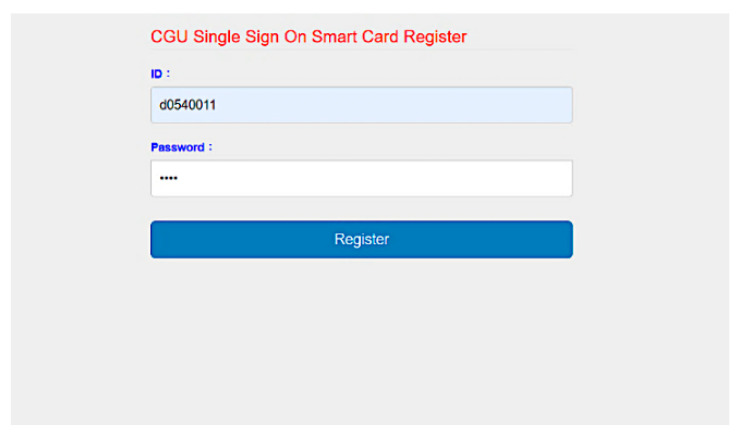
Interface of registration.

**Figure 15 sensors-21-02880-f015:**
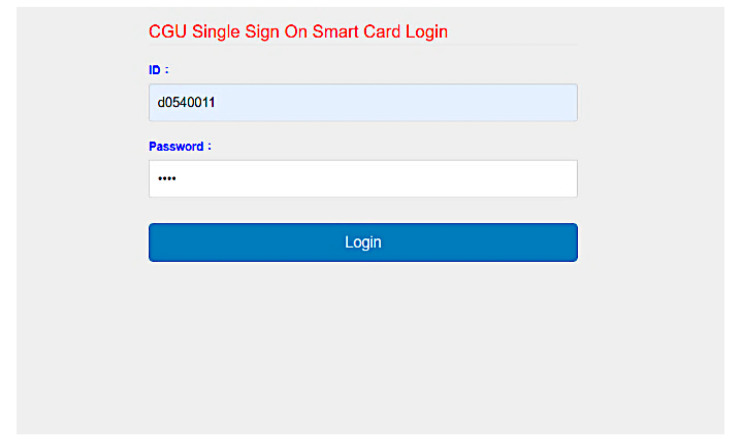
Interface of login.

**Figure 16 sensors-21-02880-f016:**
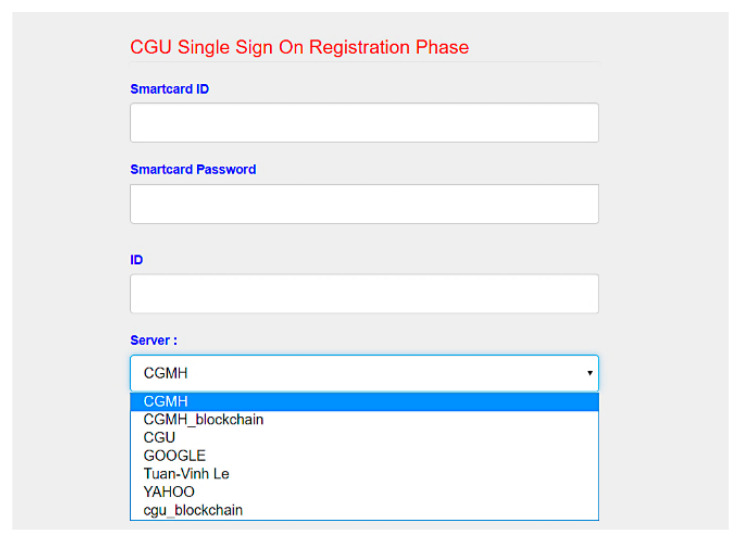
Interface of choosing services.

**Figure 17 sensors-21-02880-f017:**
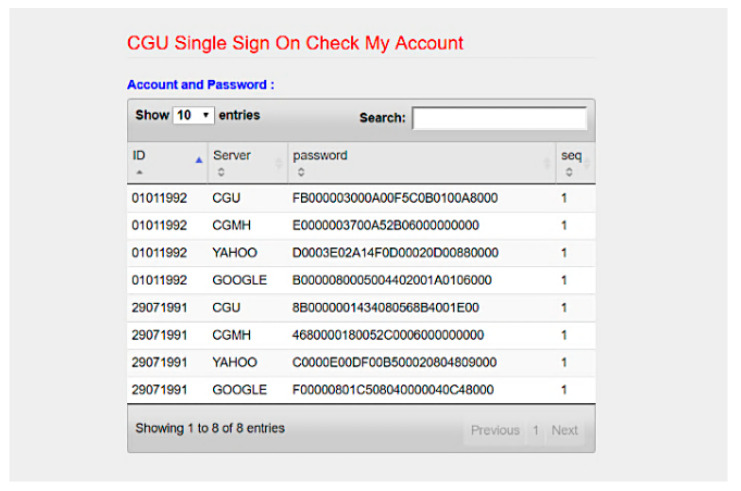
Interface of account checking.

**Table 1 sensors-21-02880-t001:** Notations of the proposed scheme.

Notations	Definitions
${\mathrm{ID}}_{i}$	Identity of user $U_{i}$.
${\mathrm{SID}}_{j}$	Identity of server $S_{j}$.
⊕	Exclusive OR (XOR) operation.
*H*(.)	Collision-resistant one-way hash function.
PWi	Password of user $U_{i}$.
$x_{S_{j}}$	Secret value of server $S_{j}$.
*k*	Encryption/decryption key *k*.
$E_{k}$(.)/$D_{k}$(.)	A symmetric encryption/decryption algorithm with secret key *k*.
*x*, yi, ρi	Random numbers.
$h_{k}$(.)	Collision-resistance secure one-way chaotic hash function.
USB	Portable USB device.
sj	Server $S_{j}$’s new chaotic random number.

**Table 2 sensors-21-02880-t002:** Notations of BAN logic [52] used in analyzing the proposed scheme.

Notations	Definitions
*P*, *Q*	Principles.
*X*, *Y*	Statements.
*r*, *w*	Readers (receivers) and writers (senders).
*K*	Encryption key.
*P believes X*	*P* believes *X*.
*P once said X*	*P* once said *X*.
*C*(*X*)	*X* is transited through communication channel *C*.
*r*(*C*)/*w*(*C*)	Readers/writers of *C*.
*P sees C*(*X*)	*P* sees *C*(*X*).
*P sees X*|*C*	*P* sees *X* via *C*.
${\left(X\right)}_{K}$	*X* is encrypted with the key *K*.
$P\overset{K}{↔}Q$	*P* and *Q* establish a secure communication channel using *K*.

**Table 3 sensors-21-02880-t003:** Comparisons of Security Requirements.

Properties	[63]	[46]	[48]	[64]	[49]	[65]	[38]	Ours
Preventing key-compromise impersonation attack	X	O	X	X	O	O	O	O
Preventing server spoofing attack	X	O	O	O	O	X	X	O
Multi-server environments	X	X	X	X	X	O	O	O
Preventing MITM attack	X	X	O	O	O	X	O	O
Stolen-verification table attack	X	O	O	O	X	X	O	O
Key confirmation	X	X	X	X	X	X	X	O
Preventing clock synchronization problem	O	X	X	O	X	O	X	O
User anonymity	X	X	O	O	O	X	O	O
Preventing denial-of-service (DoS) attack	X	X	X	O	O	X	O	O

**Table 4 sensors-21-02880-t004:** Comparisons of Computational Complexity.

Roles	Lee et al. [49]	Madhusudhan et al. [65]	Sureshkumar et al. [38]	Ours
User	${3T}_{h}{+3T}_{\mathrm{ch}}{+T}_{\mathrm{sym}}\;\approx {3T}_{h}{+525T}_{h}+2.5T_{h}\;=530.5T_{h}$	${8T}_{h}{+5T}_{\mathrm{ch}}\;\approx {8T}_{h}{+875T}_{h}\;{=883T}_{h}$	${7T}_{h}{+5T}_{\mathrm{ch}}\;\approx {7T}_{h}{+875T}_{h}\;{=882T}_{h}$	${5T}_{h}{+2T}_{\mathrm{ch}}{+2T}_{\mathrm{sym}}\;\approx {5T}_{h}{+350T}_{h}{+5T}_{h}\;{=360T}_{h}$
Server	$T_{h}{+3T}_{\mathrm{ch}}{+2T}_{\mathrm{sym}}\;\approx T_{h}{+525T}_{h}{+5T}_{h}\;=531.5T_{h}$	${5T}_{h}{+4T}_{\mathrm{ch}}\;\approx {5T}_{h}{+700T}_{h}\;{=705T}_{h}$	${3T}_{h}{+2T}_{\mathrm{ch}}\;\approx {3T}_{h}{+350T}_{h}\;{=353T}_{h}$	${4T}_{h}{+2T}_{\mathrm{ch}}{+2T}_{\mathrm{sym}}\;\approx {4T}_{h}{+350T}_{h}{+5T}_{h}\;{=359T}_{h}$
Both	${1061T}_{h}$	1588$T_{h}$	1235 $T_{h}$	719 $T_{h}$

$T_{\mathrm{ch}}$: Time for performing a Chebyshev chaotic maps operation; $T_{\mathrm{sym}}$: Time for performing a symmetry encryption operation; $T_{h}$: Time for performing a one-way hash function operation; $T_{\mathrm{ch}}\;\approx \;$175$T_{h}$; $T_{\mathrm{sym}}\;\approx \;$2.5$T_{h}$.

## Data Availability

Not applicable.

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
