# Peer review of "A Smartcard-Based User-Controlled Single Sign-On for Privacy Preservation in 5G-IoT Telemedicine Systems"

_sensors, 2021, doi:10.3390/s21082880_

Round 1
Reviewer 1 Report
In this paper, the authors propose a SC-UCSSO for 5G-IoT telemedicine systems, which can achieve some general security requirements. Their scheme establishes a secure communication channel using authenticated session keys between patients and services of telemedicine systems and allow patient access multiple telemedicine services with a pair of identity and password. The paper is readable and well-written. The protocol is explained and its security is analyzed thoroughly. However, I also have some comments.
The Section 2 does not sufficiently describe the new contribution to research. The paper should be acceptable for publication if this section is significantly ameliorated. Please improve this section by detailing the discussions on the limitations of the research, implementation practice guideline, and future studies. Furthermore, the authors should expand the Section 2. There are several works that have been published in recent years. List some suggestions:
- FAIDM for Medical Privacy Protection in 5G Telemedicine Systems TW Lin, CL Hsu - Applied Sciences, 2021 - mdpi.com;
- Internet of Things for in-home health monitoring systems: current advances, challenges and future directions NY Philip, JJPC Rodrigues, H Wang… - IEEE Journal on …, 2021;
- Trust and reputation in the internet of things: state-of-the-art and research challenges G Fortino, L Fotia, F Messina, D Rosaci… - IEEE Access, 2020
Reviewer 2 Report
This work presents the problem of the privacy preservation issue of sensitive and private transmitted information in telemedicine systems working in 5G-IoT public networks.
Biodata about health conditions is transferred to smart phone and analyzed by applications on smart phone without being transferred to other outside systems and medical professionals must be able to access the information immediately.
One aspect of security that is not analyzed nor even a brief indication is that of electromagnetic security and the possibility of external interaction. Electromagnetic interference (EMI) that can affect supporting wearable technology (heartbeat rate, quality of sleep, amount of exercise, and so on) by potential sources for EMI in wireless communication environments.
Reviewer 3 Report
The paper proposes a SC-UCSSO authentication scheme for telemedicine systems. It claims enhance security and performance compared with previous schemes.
Literature review section is weak. Section Section 2.1 provides an overview of telemdeicine and section 2.2 seems out of context.
Section 3: it is recommended to provide an overview of the proposed scheme first. This section starts with mathematical explanation of the proposed scheme. What is the proposed scheme (explain figure 2 first)?
How is the implementation applied in a real-life scenario (add details about this aspect in sections 5)?
Figure 4 can be represented in a better way (a sequence diagram first and then detailed set of equations)
It is not a good practice to provide equations as images. Is there a particular reason for doing so?
The paper needs a good review for spelling grammatical errors
Reviewer 4 Report
The paper addresses a topic in the field of Telemedicine, an area of interest today. The authors propose a new SC-UCSSO for telemedicine systems with enhances security and performance.
The paper is well organized, and the length is appropriate. The title is chosen correctly, and the abstract provide sufficient information to give a clear idea of what to expect from the paper.
The technical depth of the paper meets the requirements for a scientific article published in a quality journal.
Usually, abbreviations and acronyms are defined the first time they are used in the text. In the title, the authors use the acronym ”SC-UCSSO,” which is defined only on line 82, and I think it is helpful for readers to explain this acronym in the abstract. Also, the abbreviation” AVISPA” is defined in line 353. Please do these minor corrections.
Reviewer 5 Report
This paper aims to propose SC-UCSSO for telemedicine systems. Minor revisions are required:
- Consider changing the title of the manuscript. Don't use abbreviations in the title of the manuscript.
- Abstract - consider revising the background to reflect the specific goal of this paper. The abstract should include information about the background, aim, methods, and results.
- Add the Discussion section. Your discussion section should be constructed as follows:
- rephrase the question followed by the answer that was reached from the results;
- describe how the data support the answers to the questions;
- compare to other studies;
- present the strengths and limitations;
- combine the information in the previous paragraphs into a coherent whole, within the framework of the hypotheses.
Round 2
Reviewer 3 Report
Overall, authors addressed the comments from first round of feedback. The updated text is highlighted in green. It will be nice to have a separate file listing how the feedback is addressed.